# An Evaluation of Chinese Rapeseed Production Efficiency Based on Three-Stage DEA and Malmquist Index

**Qing Li and Cong Wang ***

School of management, Dalian Polytechnic University, Dalian 116034, China
* Correspondence: 20072095137732@xy.dlpu.edu.cn; Tel.: +86-1874-2053-986

**Abstract:** As a widely grown oil crop in China, rapeseed accounts for 38.6% of China's total oil crop production. In order to ensure the sustainable and healthy development of the rapeseed industry, this essay uses the input-output panel data of 15 central rapeseed-producing provinces (cities) in China from 2011 to 2019. The reasons for the change of the index were further studied using three-stage data envelopment analysis (DEA) and the Malmquist method. The three-stage DEA results show that the scale efficiency of rapeseed production increases in other provinces, except in Qinghai Province and Gansu Province, and that scale efficiency can be improved through additional investment. Planting enterprises can improve their efficiency by improving the efficiency of scientific and technological progress. On the other hand, government expenditure has a positive impact on rapeseed production efficiency. The Malmquist index shows that the technical progress efficiency index has the most significant impact during this period and the rapeseed total factor productivity can be improved by improving technological progress efficiency. In addition, according to the measured results, corresponding suggestions are put forward.

**Keywords:** total factor productivity; rapeseed; data envelopment analysis; efficiency evaluation

## 1. Introduction

With the development of the economy and people's living standards, people's dietary structures have undergone essential changes: while the oil demand is increasing year by year [1]. Rapeseed has a long history of planting in China. Its main product, rapeseed oil, is one of the world's three most prominent edible oils and also plays a significant role in the Chinese edible oil market. Meanwhile, rapeseed oil has a large amount of unsaturated fatty acids and vitamin E, which can prevent antioxidation and reduce the risk of cardiovascular and cerebrovascular disease. In 2017, the "State Council of China proposed the policy of focusing on superior varieties and production areas and implementing refined management, encouraging and guaranteeing the production of crops" [2]. In 2019, the output of edible vegetable oil in China was 31.34 million tons, ranking second in the world. However, China canceled the policy of the base purchase of rapeseed in the same year, which reduced the profits of rapeseed and led to an unbalance economy of planting [3,4]. At the same time, China's soybean imports to the world account for 70% of the soybean consumed by China's soybean oil, which is seriously affected by international trade. Compared with soybean, the sown area and output of rapeseed rank first among oilseed crops in China and the proportion of imported raw materials and finished products is low. On this premise, improving the efficiency of rapeseed planting and the rapeseed industry will help ensure the overall safety of the Chinese edible oil supply structure and maintain the crucial strategic significance of the self-sufficiency of rapeseed oil. However, rapeseed production depends more on the enthusiasm of the planting enterprises or farmers. After China canceled the basic purchase policy, the excitement of Chinese rapeseed planting enterprises has significantly been hit, which is not conducive to the industry's sustainable development [5].

If the rapeseed planting industry continues to develop with low income or high input and low efficiency, it will bring many problems and have a particular impact on the industry [6]. On the premise of not reducing income, how to reduce input without reducing output is the core of this essay. Therefore, this essay starts with rapeseed production efficiency and total factor productivity in China. Through the evaluation of these two kinds of efficiency, it finds out the factors that reduce planting efficiency and analyze them. Through the analysis of the results, suggestions to improve efficiency were put forward to ensure the sustainable development of rapeseed planting. The remaining structure of the essay is as follows: Section 2 discusses the research literature of previous scholars. Section 3 discusses research methods and Section 4 describes indicators and data sources, including input, output, and environmental variables. Section 5 provides the empirical results. Section 6 is the discussion and summary.

## 2. Literature Review

The production efficiency of agricultural products is the focus of the government and enterprises. Various methods of estimating the efficiency emerge endlessly in the research on the efficiency of farm products. In the study of bulk crops, the scale efficiency of agricultural products is the focus of scholars' research. Wang et al. used the DEA analysis method to analyze the relationship between the size of rice in six rice-growing counties in Nanjing and the rent paid by farmers for agricultural machinery to calculate the proportion of rent borne by farmers for farm machinery and the output-input ratio [7]. Li et al. believed that the higher the development of economies of scale, the lower the land productivity. However, the increase in land scale can reduce the agricultural production cost per unit area. Under the statistical caliber of land output per unit area, the higher the land scale, the lower the growth rate of land output per unit area [8]. Mei et al. believed there was a specific relationship between land productivity and the degree of scale. The land with low scales would lead to low land productivity [9]. When studying the production efficiency of crops, Luo et al. found that only when large agricultural machinery replaces manual labor will the yield increase [10]. The authors of [11] studied different business entities and found that family farms have high comprehensive efficiency and low production costs of cooperatives. The reason is that the staff of family farms is directly linked to the output and income of crops, so they are highly motivated to plant.

Data Envelopment Analysis (DEA) is also widely used by scholars in many fields, including agricultural production efficiency analysis. This method is a non-parametric linear programming technique that can deal with the sample efficiency analysis of multi-input and multi-output. It does not need to assume the form of a production function or the relevant parameters in advance. Hong et al. used the three-stage DEA model to evaluate the efficiency of the local government debt and put forward constructive suggestions [12]. Zhang et al. used the DEA model to evaluate the efficiency of green technology innovation in industrial enterprises [13]. The BCC model analysis of the three-stage DEA model can also be used to prove the feasibility of rural credit cooperative reform [14]; Tian et al. used the super-efficiency DEA model to calculate the efficiency of agricultural modernization development [15]. The authors of [16] used the DEA-Tobit model to analyze the efficiency of forestry ecological security and its influencing factors. Wang and Chen used a two-stage DEA model to evaluate the impact of university scientific and technological innovation on regional innovation performance and pointed out the factors that affect efficiency [17]. Sun et al. adopted the DEA cross-efficiency model to evaluate the economic benefits of urban public infrastructure [18]. By analyzing and comparing the cost–benefit efficiency of rapeseed in China and Canada, Li et al. concluded that the production cost of rapeseed in Canada is high, the quality deviation is high, and the farmers' profit from planting rapeseed is low [19]. Scholars have also used data envelopment analysis models to measure the agricultural commodities' total factor productivity (TFP). For example, scholars use the stochastic frontier production function model to calculate soybean TFP. The Malmquist-Luenberger study on soybean TFP under carbon emission or the Hicks-Moorsteen method

was used to analyze the TFP in Western and Northeast China [20,21]. Li et al. studied the spatiotemporal heterogeneity of TFP in wheat and maize [22]; Chen et al. analyzed the TFP of oil crops from the perspective of the disaster rate and found that drought was the main factor affecting the output efficiency of oil crops [23]. Zhu et al. calculated the impact of agricultural infrastructure on the TFP of major food crops [24]. The authors of [25] analyzed soybean TFP in the main soybean producing provinces and pointed out the factors affecting soybean TFP. Li et al. conducted empirical measurements on panel data of 13 major rapeseed-producing areas in China and found that rapeseed production in China experienced a net outflow of labor [26]. The authors of [27] found the root cause of the decline in the competitiveness of the soybean industry through the horizontal comparison between the soybean and its competitive crop corn.

Data Envelopment Analysis (DEA model) is a non-parametric linear programming technique that can deal with multiple inputs and sample efficiency analysis of input and output. In the first-stage DEA model, the default management inefficiency leads to the deviation of the research objective and the efficiency frontier. It ignores the influence of the external environment and random error of the decision-making unit (DMU) on the efficiency, which greatly reduces the scientific reliability of the research results. Coelli introduced the least square linear analysis based on the one-stage DEA model. However, Coelli's DEA method did not consider the efficiency value and decreased the influence of external environmental factors and random interference on the efficiency. Under this premise, the three-stage DEA proposed by Fried can compensate for this deficiency well. It can remove the interference of environmental elements and random errors on the value and introduce stochastic frontier analysis (SFA) for regression analysis. The efficiency value obtained is the efficiency value excluding the interference of the external environment and random errors. So, the three-stage DEA based on predecessors compensates for this deficiency, which can peel off the interference of environmental factors and random errors on the efficiency value, further developing the data envelopment method. In the first stage, the traditional DEA model is applied to analyze the efficiency of the initial data. In the second stage, the slacks variables of input factors are used as dependent variables and the environmental variables are used as independent variables. The stochastic frontier analysis (SFA) is introduced for regression analysis and the initial variables are adjusted according to the regression results. In the third stage, the adjusted value replaces the initial input value. Then, the traditional DEA model is used to calculate the efficiency value, eliminating the influence of the external environment and random error interference.

This essay mainly measured the production efficiency and total factor productivity of rapeseed. We aim to find the minimum input without lowering production and maximize the enterprise's profits. To measure the actual efficiency, obtain the "real" deficiencies in production, and more accurately measure the weaknesses of each DMU (rather than being masked by external factors) and make suggestions. Therefore, this essay initially uses the three-stage DEA method for analysis to obtain the real efficiency value. After estimating the comprehensive efficiency of 15 central rapeseed-producing provinces in China from 2011 to 2019, using the DEA Malmquist model proposed by Fare in 1996, the original dynamic efficiency evolution value is obtained and the state efficiency evolution value for eliminating statistical noise are measured. After analysis, this essay concludes and proposes corresponding countermeasures [28–30].

## 3. Methods

In this essay, the comprehensive efficiency of the main rapeseed-producing areas in China is evaluated based on the three-stage DEA method proposed by Fried et al. The DEA Malmquist model is then used to evaluate the TFP [31]. In the first stage, the BCC model proposed by Banker, Charnes, and Cooper in the first stage is adopted. This model

assumes that DMU is in the case of variable returns to scale, which is used to measure pure technology and scale efficiency. The BCC formula can be expressed as Equation (1):

$$
\begin{aligned}
&\min\theta\\
&s.t.\\
&\sum_{j=1}^{n} \lambda_j x_{ij} \leq \theta x_{ik}\\
&\sum_{j=1}^{n} \lambda_j y_{rj} \geq y_{rk}\\
&\sum_{j=1}^{n} \lambda_j = 1\\
&\lambda \geq 0; i = 1, 2, \ldots, m; r = 1, 2, \ldots, q; j = 1, 2, \ldots, n;
\end{aligned}
\tag{1}
$$

where $x_{ij}$ is the $i$th quantity input to unit $j$, $y_{ij}$ is the $r$th quantity output from unit $j$, and n is the number of decision-making units (DMUs). $m$ is the number of inputs and $q$ is the number of outputs. $x_{ik}$ and $y_{rk}$ are the $i$th input and $r$th output, respectively, of the evaluated $DMU_k$. $\lambda_j$ is the $j$ dimensional weight vector of $DMU_j$. We aim to measure the extent to which the resource input can be reduced without reducing the planting output. Therefore, the constraint $\sum_{j=1}^{n} \lambda j = 1$ is added. We can calculate the TE (Technical Efficiency) value and the PTE (Pure Technical Efficiency) value according to the formula SE = TE/PTE, which can calculate the scale efficiency (SE).

In the first stage of the three-stage DEA model, from the total slack of the ith input factor pair $j$, $S_{ij}$ can be derived from the first stage BCC model. $S_{ij}$ shows the gap between the theoretical and actual input of the optimal plant production efficiency. In the second stage, the input slacks variables $Sij$ obtained from the BCC analysis in the first stage are regressed with the observable environmental variables and error terms through stochastic frontier analysis (SFA). The regression equation can be expressed as Equation (2):

$$
\begin{aligned}
s_{ij} &= f(Z_j, \beta_i) + u_{ij} + v_{ik}\\
\varepsilon_{ij} &= u_{ij} + v_{ik}
\end{aligned}
\tag{2}
$$

In Equation (3), $Z_j$ is a vector representing the $t$ environmental variables of the $j$th DMU, $Z_j = (z_{1j}, z_{2j},\ldots, z_{tj})$. $\beta_i$ is the coefficient vector of environmental variables. $(Z_j, \beta_i) = Z_j \times \beta_i$ can calculate the environmental value affecting the rapeseed production output. $\varepsilon_{ij} = u_{ij} + v_{ij}$ is the combined error term, $u_{ij}$ and $v_{ij}$ are unrelated variables, $u_{ij}$ reflects the management inefficiencies of the $j$th input and the $i$th input of $u_{ij}$, $u_{ij} \sim N^+(0, \sigma_{ui}^2)$ and $v_{ij}$ reflect the statistical noise $v_i \sim N (0, \sigma_{vi}^2)$ of the $j$th DMU and the $i$th input of $v_{ij}$. The SFA method decomposes the slacks variables into environmental impact, inefficient management, and statistical noise. The adjustment input of each DMU can be calculated by Equation (3):

$$
X^A_{ij} = x_{ij} + \left[\underset{i}{MAX}\left(Z_j \times \hat{\beta}_i\right) - Z_j \times \hat{\beta}_i\right] + \left[\underset{i}{MAX}(\hat{v}_{ij}) - \hat{v}_{ij}\right]
\tag{3}
$$

$X^A_{ij}$ and $x_{ij}$ are input values of adjustment and observation, respectively, but are estimates of $\beta_i$ obtained using SFA analysis, while $\hat{\beta}_i$ is an estimate of $\beta_i$ obtained using SFA analysis. The $\left[\underset{i}{MAX}\left(Z_j \times \hat{\beta}_i\right) - Z_j \times \hat{\beta}_i\right]$ right side of Equation (3) classifies the operating environment for all DMUs. $\left[\underset{i}{MAX}\left(\hat{v}_{ij}\right) - \hat{v}_{ij}\right]$ classifies all DMUs into something similar to "luck". To obtain an estimate of $\hat{v}_{ij}$ for each DMU, an estimate of the statistical noise residual according to Jondrow et al. is used. See Equation (4):

$$
\hat{E}\left[\underset{i}{vij}\middle|u_{ij} + v_{ij}\right] = s_{ij} - Z_i \times \beta_i - \hat{E}\left[u_{ij}\middle|u_{ij} + v_{ij}\right]
\tag{4}
$$

Then, the adjusted input can be obtained using Equation (5):

$$X^A{}_{ij} = x_{ij} + \left[ \underset{i}{MAX} \left( Z_j \times \hat{\beta}_i \right) - Z_j \times \hat{\beta}_i \right] + \left[ \underset{i}{MAX} (\hat{E}(9 \left| u_{ij} + v_{iji} \right]) - \hat{E} \left[ u_{ij} \middle| u_{ij} + v_{ij} \right]_i \quad (5)$$

In the third stage of the DEA test, the BCC model is executed again using the adjusted input data to evaluate the "real" productivity.

The decision-making unit (DMU) includes cross-section data and time series data. In this case, using traditional DEA models (CCR model, BCC model) to evaluate the panel data would contradict DEA assumptions. However, the Malmquist exponential model can measure the dynamic efficiency of the time series data. After completing the first three stages of testing, the Malmquist index test is conducted on the input and output before and after the adjustment and the Farrell distance function is quoted. The DEA-Malmquist index, which uses period $t$ as the base period and is output oriented, can be expressed as (6):

$$M_i^t = \frac{D_i^t \left( x^{t+1}, y^{t+1} \right)}{D_i^t (x^t, y^t)} \quad (6)$$

Similarly, the Malmquist index with $t + 1$ period as the base period is (7):

$$M_0 \left( x^{t+1} y^{t+1}, x^t, y^t \right) = \left[ \left( \frac{D_0^t \left( x^{t+1}, y^{t+1} \right)}{D_0^t (x^t, y^t)} \right) \left( \frac{D_0^{t+1} \left( x^{t+1}, y^{t+1} \right)}{D_0^{t+1} (x^t, y^t)} \right) \right]^{1/2} \quad (7)$$

where, $D_0^t (x^t, y^t)$ and $D_0^t (x^{t+1}, y^{t+1})$ represent the output distance function of time $t$ and time $t + 1$ with time $t$ as the reference point. Similarly, $D_0^{t+1} (x^{t+1}, y^{t+1})$ and $D_0^{t+1} (x^t, y^t)$ also represent the output distance function of time $t$ and time $t + 1$ with the front technology as reference at time $t + 1$.

Most production entities usually have a certain degree of technical inefficiency. Therefore, the productivity improvement in the Malmquist index results from the change in comprehensive and technical efficiency. The comprehensive efficiency refers to the maximum production capacity of a production organization under specified conditions. The closeness measures efficiency to the production boundary, reflecting the production organization's effective use of technology. The technological progress leads to the expansion of productivity; that is, productivity changes over time. On the premise of constant returns to scale, Formula (8) can be used to deduce:

$$M_0 x^{t+1}, y^{t+1}, x^t, y^t$$
$$= \left[ \left( \frac{D_0^t \left( x^{t+1}, y^{t+1} \right)}{D_0^t (x^t, y^t)} \right) \times \left( \frac{D_0^t \left( x^{t+1}, y^{t+1} \right)}{D_0^{t+1} \left( x^{t+1}, y^{t+1} \right)} \times \frac{D_0^t \left( x^t, y^t \right)}{D_0^{t+1} (x^t, y^t)} \right) \right]^{1/2} \quad (8)$$

At this time, the two parts of the comprehensive efficiency and technological progress efficiency will represent total factor productivity. However, in actual production, the return on the scale reward is not constant, so equation Formula (9) can be reduced and derived as follows:

$$M_0 x^t, y^t, x^{t+1}, y^{t+1}$$
$$= \frac{S_0^{t+1} \left( x^{t+1}, y^{t+1} \right)}{S_0^t (x^t, y^t)} \times \frac{D_0^{t+1} \left( x^{t+1}, \frac{y^{t+1}}{VRS} \right)}{D_0^t \left( x^t, \frac{y^t}{VSR} \right)} \times \left[ \frac{D_0^t \left( x^{t+1}, y^{t+1} \right)}{D_0^{t+1} \left( x^{t+1}, y^{t+1} \right)} \times \frac{D_0^t \left( x^t, y^t \right)}{D_0^{t+1} (x^t, y^t)} \right]^{1/2} \quad (9)$$

In the Malmquist index, the comprehensive efficiency can be divided into pure technical efficiency and scale efficiency. The pure technical efficiency represents the technical efficiency level of the year after removing the scale efficiency. If the pure technical efficiency is less than 1, the invested resources have yet to play a role at the current technological level. The scale efficiency can measure the distance between different production fronts

under the two conditions of constant scale returns and variable scale returns, supposing the scale efficiency is less than 1. It indicates that the resources invested in the technical level of the model can still be increased and the resources in terms of scale can be increased to improve the scale efficiency.

## 4. Data

### 4.1. Selection of Indicators

In this study, 15 rapeseed production regions (provinces) in China were selected as the decision-making units of the three-stage DEA model and the Malmquist model. As the most original and direct production mode, crop input and output indicators are more straightforward than other manufacturing (industrial) products. So, this essay starts with the production method, selects the primary input-output data of rapeseed production, and combines the results of previous scholars to conduct research. This essay's input, output, and environmental variables data are from the China Statistical Yearbook, Cost Benefit Collection of Chinese Agricultural Products, and China Rural Yearbook. Table 1 shows the indicator system, including input, output, and environmental variables. See Table 1 for details:

**Table 1.** Input-output indicators.

| Input Indicators | Output Indicators | Environmental Variables |
|---|---|---|
| Material and service expenses | Output of main products | Regional GDP |
| Discount for domestic labor | | Education level in rural areas |
| Employee expenses | | Government financial expenditure |
| Rent of circulating land | | Urbanization level |
| Discount lease of self-operated land | | |

### 4.2. Input Indicators

The input in this study is defined as the cost of resources, materials, and services for rapeseed production. Household labor discounts, employee fees, and land rents are the most basic and most used inputs in agricultural production. In the production of farm products, the most important is the cost of seeds and fertilizer. The second one is agricultural machinery, hydraulic irrigation, and other expenses [32]. In this essay, the above fees are included in the material and input costs. The labor cost is divided into household labor discounts and employee costs. At this stage, many farmers include their labor in the input cost. During the busy farming season, employees are engaged in activities such as sowing and harvesting, so the cost of employees is also a significant expenditure. In the cost input of land, we select the land lease expense and self-operated land discount as the input cost because renting has become the choice of most farmers [33]. Therefore, it is scientific to choose the above decision-making units as inputs.

### 4.3. Output Indicator

The selected output indicators are shown in calculating agricultural products in this essay. The ultimate significance of this essay is the production efficiency of oil crops, so only the output of the rapeseed is selected as the output indicator.

### 4.4. Environment Variable

The output and production efficiency of crops will also be affected by external factors, such as regional GDP, rural education, and the urbanization level of government financial expenditure. The regional GDP has a particular impact on the production of agricultural products. The higher the provincial GDP is, the more developed the economy is. The farmers' incomes, the inputs during planting, the labor costs during planting, and the input costs of goods will also increase. At the same time, the value of agricultural products received will also increase. The education level in rural areas will affect the planting

technology of agricultural producers, the acceptance of modern technologies, and the proficiency of production machinery. It will impact planting technology and strategies [34–37]. Policies and subsidies can positively impact rapeseed production in various ways, so it is reasonable to include government expenditure in environmental factors. In agricultural production areas, the improvement of urbanization level will significantly enrich the access to agricultural materials and promote the spread of planting technology. On the contrary, the higher the degree of urbanization, the more it will attract rural labor to transfer to cities and towns, leading to the loss of rural labor to a certain extent, increasing the labor cost and price cost of crop production. Therefore, it is reasonable to include the level of urbanization in the environmental variables [38].

## 5. Results

### 5.1. The First Stage: DEA Model Results

We can observe from Table 2 that the first stage DEA indicates that the average comprehensive efficiency of the 15 central production provinces in 2011–2019 is approximately 0.85 in most years. The highest year is 2018, which is 0.982, and the lowest is 2019, which is 0.552. The increase and decrease in these two years are significant. The provinces with positive scale efficiency performance include Inner Mongolia, Anhui, and Jiangxi. The efficiency has been improved in recent years. The Yunnan Province, Guizhou Province, Chongqing City, and Hunan Province, the efficiency of these four planting regions is always low and the comprehensive efficiency has yet to reach the optimal value. According to the analysis data, we can improve the total efficiency from scale and pure technical efficiency. However, in most years, Jiangsu, Gansu, and Qinghai are always in a state of diminishing returns to scale (DRS). It indicates that the input and output are always mismatched. The input can be appropriately reduced. To achieve the best efficiency value. We can see the results in Table 2:

**Table 2.** DEA analysis results of the first stage of Chinese rapeseed output.

| Province/Year | 2011 | | | 2012 | | | 2013 | | |
|---|---|---|---|---|---|---|---|---|---|
| | CE | PTE | SE | CE | PTE | SE | CE | PTE | SE |
| Average | 0.803 | 0.856 | 0.937 | 0.722 | 0.785 | 0.919 | 0.78 | 0.798 | 0.978 |
| Inner Mongolia | 0.332 | 1 | 0.332 | 0.444 | 1 | 0.444 | 0.419 | 1 | 0.419 |
| Jiangsu | 0.906 | 1 | 0.906 | 0.655 | 0.689 | 0.95 | 0.8 | 1 | 0.8 |
| Zhejiang | 0.971 | 1 | 0.971 | 1 | 1 | 1 | 0.917 | 0.931 | 0.984 |
| Anhui | 0.772 | 0.856 | 0.902 | 0.636 | 0.804 | 0.791 | 0.957 | 1 | 0.957 |
| Jiangxi | 0.823 | 1 | 0.823 | 0.76 | 0.869 | 0.875 | 1 | 1 | 1 |
| Henan | 0.854 | 1 | 0.854 | 0.758 | 1 | 0.758 | 0.961 | 1 | 0.961 |
| Hubei | 1 | 1 | 1 | 0.94 | 0.976 | 0.963 | 1 | 1 | 1 |
| Hunan | 0.939 | 1 | 0.939 | 0.859 | 0.907 | 0.948 | 0.832 | 0.834 | 0.998 |
| Chongqing | 0.566 | 0.661 | 0.856 | 0.576 | 0.642 | 0.898 | 0.562 | 0.608 | 0.924 |
| Sichuan | 0.88 | 0.883 | 0.996 | 0.73 | 1 | 0.73 | 0.789 | 0.809 | 0.975 |
| Guizhou | 0.626 | 0.846 | 0.74 | 0.542 | 0.667 | 0.813 | 0.838 | 0.923 | 0.907 |
| Yunnan | 0.473 | 0.604 | 0.784 | 0.655 | 1 | 0.655 | 0.466 | 0.565 | 0.825 |
| Shaanxi | 0.615 | 0.662 | 0.928 | 0.487 | 0.519 | 0.938 | 1 | 1 | 1 |
| Gansu | 0.795 | 0.817 | 0.973 | 0.64 | 0.678 | 0.944 | 0.567 | 0.646 | 0.877 |
| Qinghai | 1 | 1 | 1 | 0.675 | 0.813 | 0.83 | 0.931 | 1 | 0.931 |
| Province/Year | 2014 | | | 2015 | | | 2016 | | |
| | CE | PTE | SE | CE | PTE | SE | CE | PTE | SE |
| Average | 0.838 | 0.839 | 0.998 | 0.84 | 0.854 | 0.984 | 0.865 | 0.873 | 0.99 |
| Inner Mongolia | 0.401 | 1 | 0.401 | 0.483 | 1 | 0.483 | 0.591 | 1 | 0.591 |
| Jiangsu | 0.889 | 1 | 0.889 | 0.921 | 1 | 0.921 | 0.885 | 1 | 0.885 |

**Table 2.** *Cont.*

| Province/Year | | | | | | | | | |
|---|---|---|---|---|---|---|---|---|---|
| Zhejiang | 0.978 | 0.98 | 0.998 | 0.882 | 0.895 | 0.985 | 0.791 | 0.791 | 1 |
| Anhui | 1 | 1 | 1 | 1 | 1 | 1 | 0.977 | 1 | 0.977 |
| Jiangxi | 1 | 1 | 1 | 1 | 1 | 1 | 0.959 | 1 | 0.959 |
| Henan | 0.962 | 1 | 0.962 | 0.967 | 1 | 0.967 | 1 | 1 | 1 |
| Hubei | 0.986 | 0.991 | 0.995 | 0.919 | 0.933 | 0.985 | 1 | 1 | 1 |
| Hunan | 0.793 | 0.891 | 0.89 | 0.777 | 0.811 | 0.958 | 0.716 | 0.816 | 0.877 |
| Chongqing | 0.625 | 0.72 | 0.869 | 0.615 | 0.647 | 0.949 | 0.723 | 0.747 | 0.968 |
| Sichuan | 0.968 | 1 | 0.968 | 0.975 | 1 | 0.975 | 1 | 1 | 1 |
| Guizhou | 0.717 | 0.756 | 0.948 | 0.801 | 0.803 | 0.997 | 0.797 | 0.841 | 0.947 |
| Yunnan | 0.646 | 0.65 | 0.995 | 0.714 | 0.75 | 0.952 | 0.808 | 0.874 | 0.924 |
| Shaanxi | 0.979 | 0.997 | 0.982 | 1 | 1 | 1 | 1 | 1 | 1 |
| Gansu | 0.795 | 0.823 | 0.965 | 0.784 | 0.829 | 0.946 | 0.825 | 0.856 | 0.964 |
| Qinghai | 0.945 | 1 | 0.945 | 0.835 | 0.916 | 0.911 | 0.983 | 0.99 | 0.993 |

| Province/Year | 2017 | | | 2018 | | | 2019 | | |
|---|---|---|---|---|---|---|---|---|---|
| | CE | PTE | SE | CE | PTE | SE | CE | PTE | SE |
| Average | 0.851 | 0.853 | 0.998 | 0.982 | 0.995 | 0.987 | 0.522 | 0.546 | 0.956 |
| Inner Mongolia | 0.691 | 1 | 0.691 | 1 | 1 | 1 | 1 | 1 | 1 |
| Jiangsu | 0.951 | 1 | 0.951 | 0.896 | 1 | 0.896 | 1 | 1 | 1 |
| Zhejiang | 0.92 | 0.93 | 0.989 | 0.93 | 1 | 0.93 | 0.44 | 0.477 | 0.922 |
| Anhui | 0.984 | 1 | 0.984 | 1 | 1 | 1 | 1 | 1 | 1 |
| Jiangxi | 1 | 1 | 1 | 1 | 1 | 1 | 0.557 | 0.575 | 0.969 |
| Henan | 1 | 1 | 1 | 0.79 | 0.825 | 0.957 | 0.585 | 0.666 | 0.877 |
| Hubei | 1 | 1 | 1 | 0.971 | 1 | 0.971 | 1 | 1 | 1 |
| Hunan | 0.732 | 0.805 | 0.91 | 0.907 | 0.939 | 0.966 | 0.847 | 1 | 0.847 |
| Chongqing | 0.662 | 0.676 | 0.98 | 0.713 | 0.775 | 0.919 | 0.649 | 0.799 | 0.812 |
| Sichuan | 0.944 | 1 | 0.944 | 1 | 1 | 1 | 0.414 | 0.454 | 0.911 |
| Guizhou | 0.717 | 0.763 | 0.94 | 0.834 | 0.93 | 0.897 | 0.686 | 0.924 | 0.742 |
| Yunnan | 0.779 | 0.84 | 0.927 | 0.689 | 0.765 | 0.901 | 0.557 | 0.579 | 0.962 |
| Shaanxi | 1 | 1 | 1 | 1 | 1 | 1 | 1 | 1 | 1 |
| Gansu | 0.776 | 0.784 | 0.989 | 0.668 | 0.713 | 0.938 | 0.714 | 0.769 | 0.929 |
| Qinghai | 0.878 | 0.925 | 0.949 | 0.953 | 0.972 | 0.98 | 1 | 1 | 1 |

CE is comprehensive efficiency, PTE is pure technical efficiency, SE is scale efficiency.

*5.2. Second Stage: SFA Analysis*

In the second stage, the SFA regresses the four environmental variables of the province's GDP, rural education level, government expenditure and urbanization level. We can tell from the data that the likelihood ratio test values of most of the four input SFA regressions are significant at a confidence level below 5%, which proves the accuracy of the one-sided error test term and the rationality of the random frontier specification. The four SFA regressions are material and service costs, household labor discounts, land transfer rents, and self-operated land discounts γ. The values are close to 1, indicating that the investment slacks are mainly due to low management efficiency rather than statistical noise. The value of the employee expense cost is 0.38, with a significant difference from 1, indicating that it is affected by statistical noise.

The regression coefficient of regional GDP is negative on input slacks and the confidence is significant at 1%. It shows that the main impact of regional GDP on the rapeseed production industry is labor cost. Additionally, the higher the regional GDP is, the more adverse to rapeseed production it is. The regression coefficient of the rural education level is nearly positive regarding material and service costs, while the rest is harmful, which is inconsistent with the expectations. When the regression coefficient is positive, the value is low and can be almost ignored and may be affected by statistical noise. The regression coefficients of government expenditure are all positive. However, the confidence level could be higher. However, it also shows that government subsidies to agriculture will improve rapeseed output efficiency. The value indicates that the slack variable of the urban population has a promoting effect.

These results proved that the selected environmental factors significantly impacted on rapeseed production efficiency. Therefore, to eliminate the effects of environmental and statistical errors on efficiency values, we must adjust variables and re-evaluate them. We can see the SFA results in Table 3:

**Table 3.** SFA regression results of the second stage.

| Variable | Material and Service Expenses | Discount for Domestic Labor | Employee Expenses | Rent of Circulating Land | Discount Lease of Self-Operated Land |
|---|---|---|---|---|---|
| Constant term | 10.656671 (9.14) * | 1.91 (18.93) * | 2.70 (6.61) * | −1.85 (0.89) ** | 4.48 (1.19) *** |
| Regional GDP | −0.00266 (0.47) * | −0.0124 (0.0017) *** | −0.3 (0.021) *** | 0.06 (0.01) *** | −0.2 (0.03) *** |
| Education level in rural areas | 0.0007 (0.023) * | −0.00187 (0.05) * | −0.006 (0.01) * | −0.003 (0.04) ** | −0.019 (0.01) * |
| Government expenditure | 0.16795 (2.68) * | 0.11 (0.06) ** | 0.3 (0.02) *** | 0.03 (0.01) *** | 0.04 (0.02) ** |
| Urbanization level | −0.05234 (0.01) *** | 0.007 (0.02) * | 0.001 (0.007) * | 0.1 (0.02) *** | −0.03 (0.009) * |
| σ2 | 2186.9709 (1809.28) * | 1658.08 (2868.43) * | 148.65 (209.03) * | 99.44 (105.76) * | 2301.85 (1100.72) ** |
| γ | 0.8424 (0.15) *** | 0.732 (0.303) ** | 0.38 (0.93) * | 0.85 (0.19) *** | 0.91 (0.08) *** |
| Log | −260.88 | 304.93 | 222.91 | 167.3 | 245.95 |
| LR test of the one-sided error | 0.66 | 0.79 | 0.36 | 0.44 | 1.41 |

Note: *, ** additionally, and *** are significant at 10%, 5%, and 1% levels, respectively.

### 5.3. Third Stage: DEA Model Results after Adjusting Input

In the third stage, the input redundancy data adjusted by the second-order SFA segment are used and then the BCC model is used to evaluate the rapeseed production efficiency. We consider the impact of environmental factors and statistical noise and the test puts all the provinces into a fair competition environment. See Table 4 for details of the results:

**Table 4.** Results of the third phase of DEA analysis.

| Province/Year | 2011 | | | 2012 | | | 2013 | | |
|---|---|---|---|---|---|---|---|---|---|
| | CE | PTE | SE | CE | PTE | SE | CE | PTE | SE |
| Average | 0.685 | 0.776 | 0.884 | 0.782 | 0.812 | 0.963 | 0.781 | 0.829 | 0.942 |
| Inner Mongolia | 1 | 1 | 1 | 1 | 1 | 1 | 0.611 | 1 | 0.611 |
| Jiangsu | 0.813 | 1 | 0.813 | 0.95 | 1 | 0.95 | 0.966 | 1 | 0.966 |
| Zhejiang | 0.869 | 1 | 0.869 | 0.952 | 0.967 | 0.984 | 0.914 | 0.945 | 0.967 |
| Anhui | 1 | 1 | 1 | 0.969 | 0.971 | 0.998 | 1 | 1 | 1 |
| Jiangxi | 0.659 | 0.897 | 0.735 | 0.665 | 0.747 | 0.891 | 0.729 | 0.82 | 0.889 |
| Henan | 0.868 | 0.874 | 0.993 | 0.908 | 0.948 | 0.958 | 1 | 1 | 1 |
| Hubei | 0.977 | 1 | 0.977 | 1 | 1 | 1 | 0.958 | 0.977 | 0.981 |
| Hunan | 1 | 1 | 1 | 0.732 | 0.808 | 0.906 | 0.656 | 0.826 | 0.794 |
| Chongqing | 0.382 | 0.382 | 0.999 | 0.492 | 0.546 | 0.902 | 0.494 | 0.563 | 0.879 |
| Sichuan | 0.849 | 0.861 | 0.986 | 0.719 | 0.723 | 0.994 | 0.741 | 0.745 | 0.994 |
| Guizhou | 0.47 | 0.471 | 0.998 | 0.586 | 0.64 | 0.916 | 0.575 | 0.682 | 0.843 |
| Yunnan | 0.638 | 0.64 | 0.997 | 0.734 | 0.749 | 0.979 | 0.753 | 0.764 | 0.985 |
| Shaanxi | 0.484 | 0.491 | 0.985 | 0.557 | 0.56 | 0.994 | 0.565 | 0.566 | 1 |
| Gansu | 0.687 | 0.689 | 0.997 | 0.62 | 0.684 | 0.907 | 0.721 | 0.761 | 0.947 |
| Qinghai | 1 | 1 | 1 | 0.734 | 0.74 | 0.993 | 0.704 | 0.712 | 0.988 |

| Province/Year | 2014 | | | 2015 | | | 2016 | | |
|---|---|---|---|---|---|---|---|---|---|
| | CE | PTE | SE | CE | PTE | SE | CE | PTE | SE |
| Average | 0.776 | 0.801 | 0.969 | 0.737 | 0.808 | 0.913 | 0.807 | 0.851 | 0.948 |
| Inner Mongolia | 0.562 | 1 | 0.562 | 0.409 | 1 | 0.409 | 0.343 | 0.854 | 0.402 |
| Jiangsu | 0.92 | 1 | 0.92 | 0.901 | 1 | 0.901 | 0.949 | 1 | 0.949 |
| Zhejiang | 0.873 | 0.985 | 0.887 | 0.933 | 0.969 | 0.964 | 0.995 | 0.995 | 1 |
| Anhui | 1 | 1 | 1 | 1 | 1 | 1 | 1 | 1 | 1 |
| Jiangxi | 0.687 | 0.828 | 0.83 | 0.686 | 0.814 | 0.842 | 0.788 | 0.864 | 0.912 |

**Table 4.** *Cont.*

| Province/Year | CE | PTE | SE | CE | PTE | SE | CE | PTE | SE |
|---|---|---|---|---|---|---|---|---|---|
| Henan | 1 | 1 | 1 | 1 | 1 | 1 | 1 | 1 | 1 |
| Hubei | 1 | 1 | 1 | 0.816 | 0.897 | 0.91 | 0.943 | 0.968 | 0.974 |
| Hunan | 0.595 | 0.768 | 0.775 | 0.654 | 0.813 | 0.804 | 0.666 | 0.775 | 0.859 |
| Chongqing | 0.521 | 0.549 | 0.95 | 0.448 | 0.559 | 0.801 | 0.464 | 0.547 | 0.848 |
| Sichuan | 0.762 | 0.789 | 0.966 | 0.708 | 0.709 | 0.998 | 0.787 | 0.809 | 0.973 |
| Guizhou | 0.622 | 0.673 | 0.925 | 0.594 | 0.68 | 0.873 | 0.668 | 0.753 | 0.888 |
| Yunnan | 0.769 | 0.801 | 0.961 | 0.652 | 0.671 | 0.972 | 0.71 | 0.728 | 0.975 |
| Shaanxi | 0.618 | 0.629 | 0.983 | 0.479 | 0.517 | 0.925 | 0.579 | 0.591 | 0.98 |
| Gansu | 0.796 | 0.803 | 0.992 | 0.765 | 0.777 | 0.985 | 0.823 | 0.842 | 0.977 |
| Qinghai | 0.769 | 0.788 | 0.976 | 0.673 | 0.695 | 0.967 | 0.852 | 0.884 | 0.963 |

| Province/Year | 2017 | | | 2018 | | | 2019 | | |
| | CE | PTE | SE | CE | PTE | SE | CE | PTE | SE |
|---|---|---|---|---|---|---|---|---|---|
| Average | 0.747 | 0.829 | 0.901 | 0.776 | 0.852 | 0.91 | 0.801 | 0.906 | 0.885 |
| Inner Mongolia | 0.355 | 0.954 | 0.372 | 0.688 | 1 | 0.688 | 0.308 | 0.972 | 0.317 |
| Jiangsu | 0.86 | 1 | 0.86 | 0.75 | 0.794 | 0.944 | 0.941 | 1 | 0.941 |
| Zhejiang | 0.885 | 0.955 | 0.927 | 1 | 1 | 1 | 0.874 | 0.972 | 0.899 |
| Anhui | 0.961 | 0.991 | 0.97 | 0.706 | 0.914 | 0.772 | 0.84 | 0.97 | 0.866 |
| Jiangxi | 0.729 | 0.869 | 0.839 | 0.745 | 0.865 | 0.862 | 0.619 | 0.917 | 0.675 |
| Henan | 0.916 | 1 | 0.916 | 0.896 | 1 | 0.896 | 0.796 | 1 | 0.796 |
| Hubei | 1 | 1 | 1 | 0.936 | 0.977 | 0.959 | 1 | 1 | 1 |
| Hunan | 0.652 | 0.773 | 0.843 | 0.785 | 0.834 | 0.942 | 0.739 | 0.876 | 0.844 |
| Chongqing | 0.474 | 0.586 | 0.808 | 0.564 | 0.635 | 0.887 | 0.558 | 0.682 | 0.819 |
| Sichuan | 0.65 | 0.722 | 0.901 | 0.777 | 1 | 0.777 | 0.827 | 0.842 | 0.981 |
| Guizhou | 0.553 | 0.72 | 0.769 | 0.565 | 0.711 | 0.795 | 0.548 | 0.826 | 0.663 |
| Yunnan | 0.488 | 0.665 | 0.734 | 0.68 | 1 | 0.68 | 0.558 | 0.782 | 0.714 |
| Shaanxi | 0.58 | 0.605 | 0.959 | 0.58 | 0.623 | 0.931 | 0.666 | 0.772 | 0.863 |
| Gansu | 0.646 | 0.74 | 0.874 | 0.731 | 0.779 | 0.938 | 0.892 | 0.909 | 0.981 |
| Qinghai | 0.689 | 0.71 | 0.97 | 0.769 | 0.891 | 0.863 | 0.874 | 0.925 | 0.945 |

After adjustment, the average of pure technical efficiency and combined efficiency of each province fell, which means that the influence of environmental factors and statistical noise led to the overestimation of the "true efficiency." The average value of the pure technical efficiency in the recent nine years fell to 0.8 and the annual average value of comprehensive efficiency is also around 0.75. This result proves that the environment factors the efficiency and reflects the accuracy of the second stage test. We know from the "true efficiency" that rapeseed planting efficiency in western provinces was generally lower than in eastern provinces. It depends on the local economy, the degree of income, and the natural conditions. In the southwest-producing areas, Yunnan, Guizhou, Sichuan, and Chongqing are also relatively low in efficiency. The lowest comprehensive efficiency is even as low as 0.48 and there is a severe discrepancy between the input and output. According to the data analysis in the table, the comprehensive efficiency of each province is generally affected by the scale efficiency. We can improve the comprehensive efficiency by increasing the scale investment.

### 5.4. Analysis of Malmquist

In this Malmquist index measurement, the initial data in the first stage and the revised data in the second stage are used. The time *t* is introduced to measure the dynamic values of various efficiencies in the past nine years. The average value is taken for observation. See Tables 5 and 6 for details:

The above data reflect the analysis results of total factor productivity of major rapeseed-producing provinces in China from 2011 to 2019 and the data of various change values. The growth index reflects the efficiency of the past nine years. From the whole table, the rapeseed's average total factor productivity is 0.928. There is a 7.2% difference from the best efficiency of the total factor productivity, which is not DEA effective. Therefore, the input-output ratio has yet to reach the optimal level and there is still room for improvement.

Among them, the provinces with an average value greater than 1 are Inner Mongolia, Zhejiang, Henan, and Guizhou. The total factor productivity of other areas is less than 1. It can be seen from the above table that the most critical factor affecting the total factor productivity is the change in technical progress efficiency. All the provinces can improve the TFP by improving the efficiency of technological progress.

**Table 5.** Decomposition average of total factor productivity of major rapeseed producing provinces in China.

| Province | effch | techch | pech | sech | tfpch |
|---|---|---|---|---|---|
| Average | 0.982 | 0.944 | 0.984 | 0.998 | 0.928 |
| Inner Mongolia | 0.982 | 1.033 | 1 | 0.982 | 1.015 |
| Jiangsu | 1 | 0.922 | 1 | 1 | 0.922 |
| Zhejiang | 1.033 | 0.977 | 1.033 | 1 | 1.009 |
| Anhui | 0.979 | 0.883 | 0.99 | 0.989 | 0.865 |
| Jiangxi | 0.983 | 0.941 | 1 | 0.983 | 0.925 |
| Henan | 1.019 | 1.172 | 1.017 | 1.002 | 1.194 |
| Hubei | 1 | 0.94 | 1 | 1 | 0.94 |
| Hunan | 1 | 0.847 | 1 | 1 | 0.847 |
| Chongqing | 1 | 0.768 | 1 | 1 | 0.768 |
| Sichuan | 1.015 | 0.92 | 1.006 | 1.009 | 0.934 |
| Guizhou | 0.969 | 1.124 | 0.997 | 0.972 | 1.089 |
| Yunnan | 0.931 | 0.962 | 0.96 | 0.97 | 0.896 |
| Shaanxi | 0.962 | 0.712 | 1 | 0.962 | 0.685 |
| Gansu | 1 | 0.777 | 1 | 1 | 0.777 |
| Qinghai | 1 | 0.983 | 1 | 1 | 0.983 |

effch is technical progress index, techch is technical efficiency change index, pech is change index of pure technical efficiency, sech is scale efficiency change index, tpfch is total factor productivity change index.

**Table 6.** Average value of Malmquist index of each province after eliminating external influence.

| Province | effch | techch | pech | sech | tfpch |
|---|---|---|---|---|---|
| Average | 0.982 | 0.944 | 0.984 | 0.998 | 0.928 |
| Inner Mongolia | 0.903 | 0.818 | 1 | 0.903 | 0.738 |
| Jiangsu | 1 | 0.92 | 1 | 1 | 0.92 |
| Zhejiang | 1.035 | 0.972 | 1.036 | 1 | 1.006 |
| Anhui | 0.979 | 0.883 | 0.99 | 0.989 | 0.865 |
| Jiangxi | 0.983 | 0.92 | 1 | 0.983 | 0.904 |
| Henan | 1.019 | 0.861 | 1.017 | 1.003 | 0.878 |
| Hubei | 1 | 0.937 | 1 | 1 | 0.937 |
| Hunan | 1 | 0.838 | 1 | 1 | 0.838 |
| Chongqing | 1 | 0.797 | 1 | 1 | 0.797 |
| Sichuan | 1.024 | 0.905 | 1.015 | 1.009 | 0.927 |
| Guizhou | 0.969 | 1.131 | 0.997 | 0.972 | 1.096 |
| Yunnan | 0.931 | 0.922 | 0.962 | 0.968 | 0.859 |
| Shaanxi | 0.593 | 1.599 | 0.688 | 0.862 | 0.947 |
| Gansu | 1.015 | 0.631 | 0.966 | 1.05 | 0.64 |
| Qinghai | 0.901 | 0.951 | 0.995 | 0.905 | 0.857 |

It can be seen from Table 6 above that the input redundancy data adjusted by the second-order SFA segment have eliminated environmental noise and statistical errors. Only the Zhejiang and Guizhou provinces have a TFP greater than 1, showing an upward trend. The total factor productivity of other provinces has declined. It shows that in the natural environment, the total factor productivity of each province is far from reaching the expected value of 1 and there is much room for improvement. Among them, the most significant decline and change is still the change in the efficiency of technological progress. From the analysis, the efficiency of technological progress still drives the reduction in total factor productivity. Among the 15 provinces, only Guizhou and Shaanxi have technical progress efficiency more significant than 1 and the areas with low technological progress efficiency

can learn from these two provinces. From the perspective of changes in scale efficiency, the average value of changes in scale efficiency of each section did not reach 1, showing a downward trend. The most significant decline was in the Shaanxi Province, which was 13.8%. The average change efficiency of Sichuan and Henan Province reached the optimal level from 2011–2019 and the trend was stable. The scale efficiency investment has been relatively perfect in recent years. The current production and development of rapeseed need to change the management, which means improving the ratio of intensive industrial production, strengthening the power of controlling the land transfer, and constantly improving the construction of the advantage of the leading industry—thus forming a scale of planting.

## 6. Discussion

We can learn from the results that the measured values of the three-stage DEA and the Malmquist reflect that the production efficiency in each main production area has yet to reach the effective value. Especially in the western production areas of China, the arid northwest and the southwest with more mountains and hills, the pure technical efficiency and scale efficiency are always low. In this circumstance, in the Gansu Province in the northwest, the pure technical efficiency and scale efficiency values hover around 0.75, not reaching the DEA effective value, and there is still about 25% space of the effective value. The average growth rate of all the factors is 0.64, which is the lowest in all major production areas. There is still 36% room for the growth to be effective. It shows a declining trend year after year. Although the pure technical efficiency value and scale efficiency value of the Qinghai Province is high, its comprehensive efficiency value is always low, hovering around 0.70, with 30% growth space. Its Malmquist index value is also low in the national ranking. The distance from the practical value is 14.3%. The Yunnan Province, Chongqing City, and Guizhou Province in southwest China have very low comprehensive efficiencies and there is even a 40% gap in the DEA effectiveness. Although the Malmquist index shows that the growth rate of these three provinces is good, the low efficiency has seriously affected the growth. In this regard, we can change the production mode on the land that is not suitable for large-scale mechanized production, produce from a green and non-polluted direction, and improve the extra benefit value of products. If the economic conditions permit it, UAVs can fertilize and spread pesticides [39–41] to improve the production efficiency. For land suitable for large-scale planting, improve its scale efficiency and reduce the invalid investment. The survey found that the production efficiency in China's western areas is significantly lower than in other areas. The production areas with the highest production efficiency are all located in the southern and coastal areas of China.

Therefore, to fundamentally improve the planting efficiency of rapeseed, we should start from the direction of input and output and the direction of financial subsidies, infrastructure construction and even irrigation [42]. On the one hand, we know from the input that the fertilizers and pesticides consumed in rapeseed production and the exhaust emissions from agricultural machinery will cause environmental pollution. After analyzing the overall efficiency, we can appropriately reduce the input of production resources and increase the investment of foundations in other directions for low-efficiency areas to improve the efficiency and decline the pollution emissions simultaneously to achieve sustainable environmental development. On the other hand, by reducing production pollution and soil hardening, production can be carried out on the premise of not destroying land productivity to achieve sustainable social development. At the same time, we are pursuing product quality and efficiency to improve economic earnings while not reducing the future income to achieve sustainable development at the economic level.

## 7. Conclusions

This essay first uses a three-stage DEA model to evaluate the operating performance of major rapeseed-producing provinces in China. It then uses the Malmquist model to analyze the dynamic total factor productivity of 15 major producing regions. In this study,

we selected five items as inputs: material and service costs, discounts for household labor, employee costs, rent of circulating land, and values for self-operated land; the rapeseed output was selected as the output. This esay cites four environmental variables: regional GDP, rural education level, government financial expenditure, and urbanization. The calculated results reflect the input redundancy efficiency in rapeseed production based on the results and also show that the most significant environmental factor affecting efficiency is government financial expenditure. At the national level, one way to improve efficiency is to increase subsidies. In terms of the importance of production, this study is aimed at local large-scale planting agricultural enterprises, which can reduce costs according to the measured results and provide some suggestions for the lower enterprises with lower benefits. As a result, they have a lot of potential for efficiency gains that can be achieved by using existing resources more efficiently rather than adding to them.

This study also has some limitations: Initially, it is difficult to obtain data and only macro data can be used for measurement. Secondly, it is impossible to produce specific statistics on the waste of chemical fertilizers, pesticides, and agricultural machinery in production. We can only put forward suggestions on increasing or reducing investment (unable to achieve point-to-point exemplary management) at a macro level to maintain the sustainability of production. Thirdly, only four environmental variables are cited in the calculation, which could be more comprehensive. For future research, the unexpected output can be introduced to measure the surprising output efficiency of pollutants $CO_2$, $SO_2$, and $NO_2$. In terms of methods, the DEA model is combined with more inspection methods to conduct more comprehensive tests and increase the stability of the results.

**Author Contributions:** Conceptualization, data, C.W. and Q.L.; methodology, software, writing—original draft preparation, C.W.; writing—review and editing, C.W.; supervision, Q.L.; funding acquisition, Q.L. All authors have read and agreed to the published version of the manuscript.

**Funding:** This research was funded by the Liaoning Provincial Social Science Planning Fund Project (L18BJL002), Humanities and Social Science Research Project of Education Department of Liaoning Province (J2019011), Dalian Academy of Social Sciences Think Tank major research topics (2019dlsky046).

**Institutional Review Board Statement:** Not applicable.

**Informed Consent Statement:** Not applicable.

**Data Availability Statement:** Data available in a publicly accessible repository. The data presented in this study are openly available in China Statistical Yearbook, Cost Benefit Collection of Chinese Agricultural Products, and China Rural Yearbook.

**Conflicts of Interest:** The authors declare no conflict of interest.

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
