# Peer review of "An Evaluation of Chinese Rapeseed Production Efficiency Based on Three-Stage DEA and Malmquist Index"

_sustainability, doi:10.3390/su142315822_

Round 1

Reviewer 1 Report

comments added

The manuscript ‘An evaluation of Chinese rapeseed production efficiency based on Three-stage DEA and Malmquist index’ has been reviewed.

The work uses a three-stage DEA model to evaluate the operation performance of major rapeseed producing provinces and also analyzes the total factor productivity of each province by using Malmquist model. I appreciate the authors for their work. However, the write up of the manuscript needs to be improved to get it suitable for Sustainability. The index change is to be explained at first it appears in text. ‘Planting enterprises can improve their efficiency’, needs better discussion. Results on findings of the Malmquist index are described however this parameter could be discussed in line with technical progress efficiency index. In literature review, instead of just describing the previous studies, authors could build a rationale, for example about the land productivity and the degree of scale. Why it was focused to use the three-stage DEA method for analysis to eliminate the impact of environmental factors. Important points including low management efficiency and statistical noise could also be linked to the performance of producers. Third line in Methods, ‘To evaluate the comprehensive efficiency…’ rephrase. Improve equation formats. Why the scale reward is not constant. The last para of Methods starting from ‘Government fiscal expenditure…’ may not belong to Methods or it needs to be rephrased since its more of introduction.  In section 5.1, instead of staring from ‘this paper uses the input-oriented BCC…’authors could directly start with results of DEA analysis. Similar comments are for other tables’ results description. Text after the Table 4, ‘After adjustment, the annual…’, explain. Discussion section is to be thoroughly developed in lien with the findings of the work, other else it seems description of software outputs of acquired data. Authors also need to discuss these in relation with sustainable development of China's rapeseed production efficiency. The Conclusions and Discussion could be limited to the key findings of this investigation and not the repletion o the analyses and background do the work.

Sincerely,

Reviewer 2 Report

Although the title of manuscript is interesting, organization of the paper is rather poor. Incomplete sentences and grammatical errors render the manuscript incomprehensible. When evaluated objectively, it is concluded that the expected care is not taken in writing the article.

Reviewer 3 Report

The authors have evaluated the production efficiency of rapeseed employing 3-stage DEA and Malmquist indices. The following points can be addressed to improve the manuscript:

1. Punctuation errors in the abstract and in the entire manuscript need to be fixed (period instead of commas and semicolons in multiple places).

2. Literature review: “Authors” to be used instead of “author” for references (which are many) that have more than one author.

3. Line nos are missing. Line nos facilitate an easier reviewing procedure.

4. The assumptions of the models used in this study need to be stated.

5. Section: No period after “Data”

6. Discussion and Conclusions sections need to be separated for a better understanding

7. Many words in the entire manuscript are double-spaced. Needs thorough revision.

8. Please include some numbers and percentages for comparative analysis against the relevant results that are already known in the literature.

9. In the discussion section, the reviewer believes there is a lack of answer to the question ‘why this trend'. A more in-depth analysis will benefit the manuscript.

10. The overall grammar of the manuscript can be improved. 

Round 2

Reviewer 1 Report

Dear Editor, 

The revised version of the manuscript has been reviewed. Authors have now revised the manuscript. Thanks to the authors. It is now suitable; however, the followings could still be improved.

-The last para of Methods starting from ‘Government fiscal expenditure…’ may not belong to Methods or it needs to be rephrased since its more of introduction.

-The Conclusions could be limited to the key findings of this investigation and not the repletion of Results. 

-The last para of Methods starting from ‘Government fiscal expenditure…’ may not belong to Methods or it needs to be rephrased since its more of introduction.

Author Response

Dear reviewer:

Thank you for your comments on this paper. For your comments, I have made the following changes:

Point 1: The last para of Methods starting from ‘Government fiscal expenditure…’ may not belong to Methods or it needs to be rephrased since its more of introduction

Response 1: Dear commentator, "government expenditure..." belongs to the interpretation of data in Chapter IV, not the method in Chapter III. Its purpose is to explain the rationality of quoting it as an environment variable. Respecting your opinions, it has been revised and briefly introduced.

Point 2: The Conclusions could be limited to the key findings of this investigation and not the repletion of Results.

Response 2: Thank you for your comments, which have been further revised.

Thank you very much for your suggestions and best wishes!

Reviewer 2 Report

Although the manuscript has been improved noteworthily, it comprises some grammatical errors. For example, in the "introduction" section, the sentence "People's consumption of starchy food such as rice and wheat are decreasing year by year, while the 26 demand for oil is increasing year by year [1]." should have been revised as "People's consumption of starchy food such as rice and wheat are decreasing year by year while the 26 demand for oil is increasing year by year [1]". Therefore, I suggested strongly they have the manuscript re-checked grammatically to get rid of grammatical debugs hidden in the lines.  

Author Response

Dear reviewer:

Thank you for your comments on this paper. For your comments, I have made the following changes:

Point 1: Although the manuscript has been improved noteworthily, it comprises some grammatical errors.

Response 1: Thank you for your comments. We have consulted people with better English and have changed the grammar.